





# Fast matrix treatment of 3D radiative transfer in vegetation canopies: SPARTACUS-Vegetation 1.0

Robin J. Hogan[1,2], Tristan Quaife[2], and Renato Braghiere[2]

[1]European Centre for Medium-range Weather Forecasts, Reading, UK.
[2]Department of Meteorology, University of Reading, Reading, UK.

*Correspondence to:* Robin J. Hogan (r.j.hogan@ecmwf.int)

**Abstract.** A fast scheme is described to compute the 3D interaction of solar radiation with vegetation. The vegetation canopy is split horizontally into one clear region and one or more vegetated regions, and the two-stream equations are used for each, but with additional terms representing lateral exchange of radiation between regions that are proportional to the area of the interface between them. The resulting coupled set of ordinary differential equations is solved using the matrix-exponential method. The scheme is compared to solar Monte Carlo calculations for idealized scenes from the 'RAMI4PILPS' intercomparison project, for open forest canopies and shrublands both with and without snow on the ground. Agreement in reflectance, transmittance and canopy absorptance is excellent in both the visible and near infrared. The technique has potential application to weather and climate modelling.

## 1 Introduction

The treatment of the interaction of vegetation with solar radiation in weather and climate models varies greatly in complexity. The simplest schemes are concerned only with surface albedo and its impact on near-surface temperature forecasts, and indeed Viterbo and Betts (1999) reported a large improvement in forecasts by the ECMWF model when the use of a fixed snow albedo was modified to account for the much lower albedo that occurs when snow falls in forested areas. Much more sophisticated treatments are used in the dynamic vegetation schemes of many climate models, for which they need to calculate also the fraction of absorbed photosynthetically active radiation (faPAR). But it was reported by Loew et al. (2014) that even state-of-the-art models, when evaluated in benchmarks for which a full physical description of the vegetation was available, had worst-case albedo errors in excess of 0.3. The challenge is to represent the complex three-dimensional structure of vegetation canopies with an radiative transfer algorithm that is nonetheless computationally efficient enough to use in a global model.

Sellers (1985) took the two-stream equations used in atmospheric radiative transfer and applied them to a vegetation canopy. In this approach, the vegetation is treated as a single horizontally homogeneous layer, and a set of three coupled ordinary differential equations are solved for the direct downwelling flux and the downwelling and upwelling diffuse fluxes. If the leaves can be assumed randomly oriented then the optical depth of the layer is equal to half the leaf area index (LAI). Meador and Weaver (1980) provided an analytic solution to these equations that is still used in a number of state-of-the-art surface energy exchange schemes (e.g., Best et al., 2011). The first-order error that arises is due to the fact that vegetation canopies are not horizontally





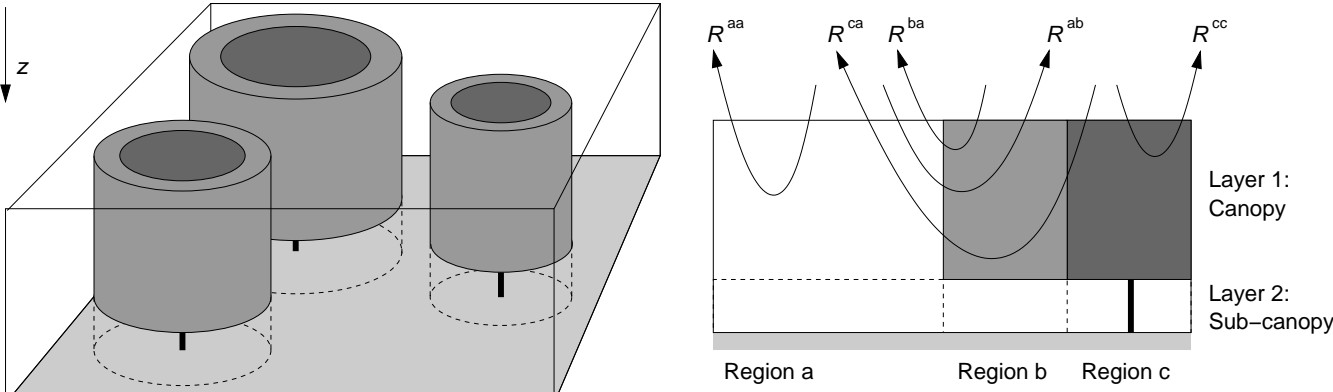

**Figure 1.** Schematic of the idealized vegetation considered in this paper, illustrating the meanings of Layers 1 and 2 and Regions $a$, $b$ and $c$. The diagram on the right also illustrates the interpretation of the elements of the reflectance matrix **R** given in (24).

homogeneous. Typically this is treated by introducing a 'clumping factor' that scales down the LAI used in the two-stream scheme. A very similar approach has previously been used in atmospheric radiation schemes to treat the clumpiness of clouds (Tiedtke, 1996). The clumping factor for vegetation is typically parameterized as an empirical function of properties of the vegetation and solar zenith angle (e.g., Ni-Meister et al., 2010), but this lacks a physical basis and fails to represent horizontal

fluxes into and out of individual vegetation elements such as trees.

Pinty et al. (2006) described one of the most sophisticated yet affordable schemes to date that attempts to overcome these limitations. Their scheme sums three terms: the reflection from the vegetation assuming a black underlying surface, the reflection from the surface assuming no interaction with the vegetation, and a term representing interactions between the surface and the vegetation. Despite much improved performance compared to the Sellers (1985) scheme, their approach still uses an empir-

ical clumping factor, and is underpinned by the Meador and Weaver (1980) solution that assumes horizontally homogeneous vegetation.

In this paper we exploit recent advances in the atmospheric literature, and adapt the 'SPARTACUS' (SPeedy Algorithm for Radiative Transfer through CloUd Sides) method of Hogan et al. (2016) to the vegetation problem. As described in section 2, this approach employs an explicit description of the horizontal distribution of vegetation for which we can write down a mod-

ified version of the two-stream equations that includes terms for lateral radiation exchange between vegetation elements and the clear regions between them. The equations are then solved exactly using the matrix-exponential method. This completely avoids the need for an effective LAI or the Meador and Weaver (1980) solution. In section 3 it is compared to Monte Carlo calculations in idealized forest and scrubland conditions.





## 2 Method

### 2.1 Overview

We use a simple geometrical description of the problem, as shown in Fig. 1. Leafy vegetation is assumed to occupy a single 'canopy layer', with the domain divided horizontally into $m$ regions, one clear region and either one or two vegetated

regions. The use of two vegetated regions adds the flexibility to represent heterogeneous tree crowns, borrowing the idea of Shonk and Hogan (2008) for representing cloud heterogeneity. While the vegetation elements are depicted in Fig. 1 as cylinders, this is not explicit assumed; rather, we assume that (1) all azimuthal orientations of the interface between the clear and vegetated regions are equally likely, and (2) the vegetation elements are randomly distributed. To represent forests, an additional 'sub-canopy layer' may be added between the vegetation and the ground, divided into the same regions as the canopy

layer above. In the case of shrubland, the sub-canopy layer may be omitted. Thus we require as a minimum just four numbers to define the geometry of the problem: the fractional area of the domain covered by vegetation, $c_v$, the vertical depth of the canopy layer, $\Delta z_1$, the vertical depth of the sub-canopy layer, $\Delta z_2$ (which may be zero), and the length of the interface between the clear and vegetated regions per unit area of the domain, $L^{ab}$. Note that although this paper considers only up to two layers and three regions, which is an appropriate level of complexity for a weather or climate model, for other applications additional

layers and regions may be added. This would enable the representation of different types of vegetation of different heights, or vegetation in the understory.

In the SPARTACUS method, the two-stream differential equations are used in each region, but with additional terms representing lateral radiation transport between regions. The formulation of these equations is given in section 2.2, with the coefficients to be used in the case of vegetation defined in section 2.3. Section 2.4 describes how they are solved for a single

layer using matrix exponentials, and section 2.5 describes the use of the adding method to compute the direct and diffuse albedos of the entire scene (vegetation and the surface beneath it). In the context of a weather or climate model, this could be done for the same spectral intervals as the atmospheric radiation scheme, or in the smaller number of broader spectral intervals for which optical properties of the vegetation and surface are defined. These albedos would then be used as boundary conditions for the calculation of the radiative flux profile in the atmosphere above. The downwelling direct and diffuse fluxes output from

the atmospheric radiation scheme are then used in section 2.6 to compute the flux profile within the vegetation canopy, enabling the absorbed and transmitted radiation to be computed. The appendix describes how the scheme may be made computationally faster by optimizing the treatment of the sub-canopy layer.

### 2.2 Differential two-stream equations in matrix form

This section summarizes the theoretical background to SPARTACUS that was introduced by Hogan et al. (2016). Solar radi-

ation in a particular spectral interval is described by three streams: the diffuse upwelling flux ($\mathbf{u}$), the diffuse downwelling flux ($\mathbf{v}$) and the direct downwelling flux ($\mathbf{s}$), where $\mathbf{u}$ and $\mathbf{v}$ are fluxes into a horizontal plane while $\mathbf{s}$ is into a plane oriented perpendicular to the sun. At any given height, these are column vectors containing the fluxes in $m$ regions; in the equations that follow we use $m = 3$ to match the schematic shown in Fig. 1, but it is straightforward to reduce to two regions. Thus





for upwelling flux we have $\mathbf{u} = \begin{pmatrix} u^a & u^b & u^c \end{pmatrix}^T$, where each flux component is defined as the radiative power divided by the area of the entire gridbox, such that the domain-mean flux is obtained by summing the elements of the vector. Optical properties of the vegetation are assumed horizontally and vertically homogeneous within each vegetated region of each layer.

The two-stream equations form a set of coupled differential equations that can be written in matrix form as

$$\frac{d}{dz}\begin{pmatrix} \mathbf{u} \\ \mathbf{v} \\ \mathbf{s} \end{pmatrix} = \mathbf{\Gamma}\begin{pmatrix} \mathbf{u} \\ \mathbf{v} \\ \mathbf{s} \end{pmatrix}, \tag{1}$$

where $z$ is height measured *downward* from the top of the layer, and $\mathbf{\Gamma}$ is a matrix describing the interactions between flux components and between different regions. It is convenient to partition it into a set of $m \times m$ component matrices as follows:

$$\mathbf{\Gamma} = \begin{pmatrix} -\mathbf{\Gamma}_1 & -\mathbf{\Gamma}_2 & -\mathbf{\Gamma}_3 \\ \mathbf{\Gamma}_2 & \mathbf{\Gamma}_1 & \mathbf{\Gamma}_4 \\ & & \mathbf{\Gamma}_0 \end{pmatrix}, \tag{2}$$

where

$$\mathbf{\Gamma}_0 = \begin{pmatrix} -\sigma_0^a/\mu_0 & & \\ & -\sigma_0^b/\mu_0 & \\ & & -\sigma_0^c/\mu_0 \end{pmatrix}$$

$$+ \begin{pmatrix} -f_{\mathrm{dir}}^{ab} & +f_{\mathrm{dir}}^{ba} & \\ +f_{\mathrm{dir}}^{ab} & -f_{\mathrm{dir}}^{ba} - f_{\mathrm{dir}}^{bc} & +f_{\mathrm{dir}}^{cb} \\ & +f_{\mathrm{dir}}^{bc} & -f_{\mathrm{dir}}^{cb} \end{pmatrix}; \tag{3}$$

$$\mathbf{\Gamma}_1 = \begin{pmatrix} -\sigma^a\gamma_1^a & & \\ & -\sigma^b\gamma_1^b & \\ & & -\sigma^c\gamma_1^c \end{pmatrix}$$

$$+ \begin{pmatrix} -f_{\mathrm{diff}}^{ab} & +f_{\mathrm{diff}}^{ba} & \\ +f_{\mathrm{diff}}^{ab} & -f_{\mathrm{diff}}^{ba} - f_{\mathrm{diff}}^{bc} & +f_{\mathrm{diff}}^{cb} \\ & +f_{\mathrm{diff}}^{bc} & -f_{\mathrm{diff}}^{cb} \end{pmatrix}; \tag{4}$$

$$\mathbf{\Gamma}_2 = \begin{pmatrix} \sigma^a\gamma_2^a & & \\ & \sigma^b\gamma_2^b & \\ & & \sigma^c\gamma_2^c \end{pmatrix}; \tag{5}$$

$$\mathbf{\Gamma}_3 = \begin{pmatrix} \sigma^a\omega^a\gamma_3^a & & \\ & \sigma^b\omega^b\gamma_3^b & \\ & & \sigma^c\omega^c\gamma_3^c \end{pmatrix}, \tag{6}$$

and $\mathbf{\Gamma}_4$ is the same as $\mathbf{\Gamma}_3$ but using the quantity $\gamma_4$ in place of $\gamma_3$. Missing entries in all these matrices are taken to be zero. The $\mathbf{\Gamma}_0$ and $\mathbf{\Gamma}_1$ matrices describe the rate at which the direct and diffuse downwelling fluxes, respectively, change along their path.





They are expressed in (3) and (4) as the sum of two matrices: the first matrix in each case represents losses due to scattering and absorption, while the second represents exchange of radiation between regions. The $\mathbf{\Gamma}_2$ matrix describes the rate of scattering of diffuse radiation from one direction to the other, while the $\mathbf{\Gamma}_3$ and $\mathbf{\Gamma}_4$ matrices describe the rate at which the direct solar beam is scattered into the upwelling and downwelling diffuse streams. The minus signs in front of the matrices on the top row

of (2) are due to this line corresponding to upwelling radiation, but the vertical coordinate increasing downward.

The symbols in (3) to (6) have the following meanings. The extinction coefficient to diffuse radiation of region $j$ is denoted $\sigma^j$, while $\sigma_0^j$ is the same but for direct radiation. The distinction between the two permits the flexibility to represent leaves with a preferred orientation. The cosine of the solar zenith angle is denoted $\mu_0$ while the single-scattering albedo is $\omega$. The coefficients $\gamma_1$–$\gamma_4$ govern the exchange of radiation between the three streams. Finally, the coefficients $f_{\mathrm{dir}}^{jk}$ and $f_{\mathrm{diff}}^{jk}$ represent

the rate at which direct and diffuse radiation, respectively, is transferred from region $j$ to region $k$. All these symbols are defined in terms of physical properties of the scene in the next section.

### 2.3 Coefficients in the two-stream equations

The matrix form of the two-stream equations in section 2.2 introduced several coefficients that are themselves functions of more fundamental optical or geometric properties. The $\gamma_1$–$\gamma_4$ coefficients may be written as (Meador and Weaver, 1980):

$$\gamma_1 = [1 - \omega(1 - \beta)]/\mu_1; \tag{7}$$

$$\gamma_2 = \omega\beta/\mu_1; \tag{8}$$

$$\gamma_3 = \beta_0; \tag{9}$$

$$\gamma_4 = 1 - \beta_0, \tag{10}$$

where $\beta$ and $\beta_0$ are the 'upscatter' fractions, the fractions of downwelling radiation (in the diffuse and direct streams respec-

20 tively) that are scattered upward, and $\mu_1$ is the cosine of the effective zenith angle of diffuse radiation. For the remainder of this paper we assume the diffuse radiation to be hemispherically isotropic, so $\mu_1 = 1/2$.

In the simplest case where leaves are assumed to be randomly oriented, the optical depth of a region is equal to half its LAI, and therefore for a layer of thickness $\Delta z$, the extinction coefficients to direct and diffuse radiation are the same and are given by

$$\sigma = \sigma_0 = \mathrm{LAI}/2\Delta z. \tag{11}$$

Assuming the leaves to be bi-Lambertian scatterers with reflectance $r$ and transmittance $t$, the single scattering albedo is given by

$$\omega = r + t, \tag{12}$$

and the upscatter fractions by

$$\beta = 1/2 + \mu_1(r - t)/3\omega; \tag{13}$$

$$\beta_0 = 1/2 + \mu_0(r - t)/3\omega. \tag{14}$$

(c) Author(s) 2017. CC BY 4.0 License.





Expressions for leaves with a preferential alignment were given by Sellers (1985).

The rates of lateral exchange of radiation between regions that appear in (3) and (4) may be derived from geometrical arguments (Hogan and Shonk, 2013; Schäfer et al., 2016) as

$$f_{\text{diff}}^{ij} = L^{ij}/2c^i; \tag{15}$$

$$f_{\text{dir}}^{ij} = L^{ij}\tan(\theta_0)/\pi c^i, \tag{16}$$

where $\theta_0$ is the solar zenith angle, $L^{ij}$ is the length of the interface between regions $i$ and $j$ per unit area of the horizontal domain, and $c^i$ is the fractional area of the domain covered by region $i$. In the $m = 3$ case we have two regions to represent horizontal heterogeneity of LAI, and following Shonk and Hogan (2008) we assume them to be of equal area, i.e. $c^b = c^c = c_v/2$ and $c^a = 1 - c_v$ (where $c_v$ is the fractional coverage of vegetation). This leads to $f_{\text{dir}}^{bc} = f_{\text{dir}}^{cb}$ and $f_{\text{diff}}^{bc} = f_{\text{diff}}^{cb}$.

The quantity $L^{ab}$ is most intuitively characterized in terms of the typical size of a vegetation element. We define the *effective tree diameter*, $D$, as the diameter of identical, cylindrical, randomly distributed and *physically separated* tree crowns in an idealized forest with the same $L^{ab}$ and tree cover $c_v$ as the real forest. In analogy to the concept of an effective cloud diameter by Jensen et al. (2008), this leads to the definition

$$L^{ab} = 4c_v/D. \tag{17}$$

If region $c$ represents the central core of the tree crowns, as depicted in Fig. 1, then this implies $L^{bc} = L^{ab}/\sqrt{2}$.

Effective tree diameter is less useful in dense forests ($c_v > 1/2$) where tree crowns touch each other leading to $L^{ab}$ decreasing with increasing tree cover. Therefore, in the more general case it is preferable to consider an *effective tree scale*, $S$, which we define such that

$$L^{ab} = 4c_v(1 - c_v)/S. \tag{18}$$

This form is inspired by the idealized geometrical analysis of Morcrette (2012). If we place idealized trees with a square footprint measuring $S \times S$ randomly on a grid, then on average the normalized perimeter length $L^{ab}$ will follow (18).

Lastly in this section, we consider how to represent the effect of vertical tree trunks in region $c$ of the sub-canopy layer (as illustrated in Fig. 1). If the trunks are of a size and number such that a horizontal slice through the sub-canopy layer intercepts a normalized total trunk perimeter (per unit area of region $c$) of $L_t$, then by analogy with (15) and (16), the diffuse and direct extinction coefficients are given by

$$\sigma = L_t/2c^c; \tag{19}$$

$$\sigma_0 = L_t\tan(\theta_0)/\pi c^c. \tag{20}$$

For simplicity we assume the trunks to be Lambertian reflectors, in which case $\omega$ is simply the trunk albedo, and with no preference for upward or downward scattering we have $\beta = \beta_0 = 1/2$.



## 2.4 Solution to equations within one layer

We may write the solution to (1) in terms of a matrix exponential (Waterman, 1981; Hogan et al., 2016): the fluxes at the base of a layer of thickness $\Delta z$ are related to the fluxes at the top of the layer via

$$
\begin{pmatrix} \mathbf{u} \\ \mathbf{v} \\ \mathbf{s} \end{pmatrix}_{z=z+\Delta z} = \exp(\mathbf{\Gamma}\Delta z) \begin{pmatrix} \mathbf{u} \\ \mathbf{v} \\ \mathbf{s} \end{pmatrix}_{z=z} , \tag{21}
$$

5  where the matrix exponential may be computed numerically using the scaling and squaring method (e.g. Higham, 2005). If 3D radiative transfer is neglected then $f_{\mathrm{diff}} = f_{\mathrm{dir}} = 0$, which decouples the equations to the extent that a computationally cheaper analytical solution is possible (Meador and Weaver, 1980). Conversely, if scattering and absorption are ignored but 3D radiative transfer is retained, a reasonable assumption in the sub-canopy layer, then $\sigma = \sigma_0 = 0$, which also decouples the equations and leads to the computationally cheaper solution given in the appendix.

10  In order to compute the flux profile, we wish to work with expressions of the following form:

$$
\mathbf{u}(z) = \mathbf{T}\mathbf{u}(z + \Delta z) + \mathbf{R}\mathbf{v}(z) + \mathbf{S}^{+}\mathbf{s}(z); \tag{22}
$$

$$
\mathbf{v}(z + \Delta z) = \mathbf{T}\mathbf{v}(z) + \mathbf{R}\mathbf{u}(z + \Delta z) + \mathbf{S}^{-}\mathbf{s}(z), \tag{23}
$$

where (22) states that the upwelling flux exiting the top of the layer is equal to transmission of the upwelling flux entering the base of the layer, plus reflection of the downwelling flux entering the top of the layer, plus scattering of the direct solar flux 15 entering the top of the layer; and similarly for (23). Figure 1 illustrates the meaning of the elements of the diffuse reflectance matrix $\mathbf{R}$ for the canopy layer:

$$
\mathbf{R} = \begin{pmatrix} R^{aa} & R^{ba} & R^{ca} \\ R^{ab} & R^{bb} & R^{cb} \\ R^{ac} & R^{bc} & R^{cc} \end{pmatrix}, \tag{24}
$$

where $R^{ij}$ is the fraction of diffuse downwelling radiation entering the top of region $i$ that is scattered out of the top of region $j$ without exiting the base of the layer. The other matrices have analogous definitions: $\mathbf{T}$ represents the transmission of diffuse 20  radiation across the layer, and $\mathbf{S}^{+}$ and $\mathbf{S}^{-}$ represent the scattering of radiation from the direct downwelling stream at the top of the layer to the diffuse upwelling stream at the top of the layer and the diffuse downwelling stream at the base of the layer, respectively.

These matrices may be derived from the matrix exponential, which we decompose into seven $m \times m$ matrices:

$$
\exp(\mathbf{\Gamma}\Delta z) = \begin{pmatrix} \mathbf{E}_{uu} & \mathbf{E}_{uv} & \mathbf{E}_{us} \\ \mathbf{E}_{vu} & \mathbf{E}_{vv} & \mathbf{E}_{vs} \\ & & \mathbf{E}_{0} \end{pmatrix} . \tag{25}
$$





It was shown by Hogan et al. (2016) that

$$\mathbf{R} = -\mathbf{E}_{uu}^{-1}\mathbf{E}_{uv}; \tag{26}$$

$$\mathbf{T} = \mathbf{E}_{vu}\mathbf{R} + \mathbf{E}_{vv}; \tag{27}$$

$$\mathbf{S}^+ = -\mathbf{E}_{uu}^{-1}\mathbf{E}_{us}; \tag{28}$$

$$\mathbf{S}^- = \mathbf{E}_{vu}\mathbf{S}^+ + \mathbf{E}_{vs}. \tag{29}$$

Moreover, the direct flux exiting the base of a layer is computed from the direct flux entering the top of a layer via $\mathbf{s}(z + \Delta z) = \mathbf{E}_0\mathbf{s}(z)$.

## 2.5 Extension to multiple layers

To compute the flux profile we use the adding method (Lacis and Hansen, 1974) but in a somewhat different form to Hogan et al.

(2016), in order to facilitate integration within a full atmospheric radiation scheme. This section considers the first part: stepping up through the vegetation layers computing the albedo of the scene below each layer interface. We define the matrix $\mathbf{A}_{i+1/2}$ as the albedo to diffuse downwelling radiation of the scene below interface $i + 1/2$ (including the surface contribution), and the matrix $\mathbf{D}_{i+1/2}$ as the albedo to direct radiation. The off-diagonal terms of these matrices represent the fraction of radiation downwelling in one region that is reflected back into the other. At the surface (interface $n + 1/2$ for an $n$-layer

description of the canopy), these matrices are diagonal:

$$\mathbf{A}_{n+1/2} = \begin{pmatrix} \alpha_{\text{diff}}^a & & \\ & \alpha_{\text{diff}}^b & \\ & & \alpha_{\text{diff}}^c \end{pmatrix}; \tag{30}$$

$$\mathbf{D}_{n+1/2} = \mu_0 \begin{pmatrix} \alpha_{\text{dir}}^a & & \\ & \alpha_{\text{dir}}^b & \\ & & \alpha_{\text{dir}}^c \end{pmatrix}, \tag{31}$$

where for maximum flexibility we allow for separate direct and diffuse surface albedos, and separate albedos below each region to represent lower snow cover beneath trees.

We then use the adding method to compute $\mathbf{A}$ and $\mathbf{D}$ just below the interface above, accounting for the possibility of multiple scattering. In the case of the diffuse albedo matrix we have

$$\mathbf{A}_{i-1/2} = \mathbf{R}_i + \mathbf{T}_i \left[\mathbf{I} + \mathbf{A}_{i+1/2}\mathbf{R}_i + (\mathbf{A}_{i+1/2}\mathbf{R}_i)^2 + \cdots\right]$$
$$\times \mathbf{A}_{i+1/2}\mathbf{T}_i, \tag{32}$$

where $\mathbf{I}$ is the $m \times m$ identity matrix. This equation states that the albedo at interface $i - 1/2$ is equal to the reflection of layer $i$,

plus the albedo at interface $i + 1/2$ accounting for the two-way transmission through the intervening layer. The term in square brackets accounts for multiple scattering between interface $i + 1/2$ and layer $i$, and since it is a geometric series of matrices,



the equation reduces to

$$\mathbf{A}_{i-1/2} = \mathbf{R}_i + \mathbf{T}_i \left(\mathbf{I} - \mathbf{A}_{i+1/2}\mathbf{R}_i\right)^{-1} \mathbf{A}_{i+1/2}\mathbf{T}_i. \tag{33}$$

Similarly, the direct albedo matrix at the interface above is given by

$$\mathbf{D}_{i-1/2} = \mathbf{S}_i^+ + \mathbf{T}_i \left(\mathbf{I} - \mathbf{A}_{i+1/2}\mathbf{R}_i\right)^{-1}$$
$$\times \left(\mathbf{D}_{i+1/2}\mathbf{E}_{0i} + \mathbf{A}_{i+1/2}\mathbf{S}_i^-\right), \tag{34}$$

where $\mathbf{D}_{i+1/2}\mathbf{E}_{0i}$ represents the direct radiation that passes down through layer $i$ without being scattered and is then reflected up from interface $i+1/2$, while $\mathbf{A}_{i+1/2}\mathbf{S}_i^-$ represents direct radiation that is scattered into the downward diffuse stream in layer $i$ and then reflected up from interface $i+1/2$. For the two-layer description of the vegetation shown in Fig. 1, (33) and (34) are applied first at interface 1.5 (between the canopy and the sub-canopy layers) and then at interface 0.5 (the top of the canopy). It is straightforward to add additional layers.

At this point we are able to compute the scalar 'scene albedos' of the surface and the vegetation. Denoting $\mathbf{c} = \begin{pmatrix} c^a & c^b & c^c \end{pmatrix}^T$ as a column vector containing the area fractions of each region, the scene albedos to diffuse and direct radiation are

$$\alpha_{\mathrm{diff,scene}} = \mathbf{c}^T\mathbf{A}_{1/2}\mathbf{c}; \tag{35}$$
$$\alpha_{\mathrm{dir,scene}} = \mathbf{c}^T\mathbf{D}_{1/2}\mathbf{c}. \tag{36}$$

When implementing the scheme described in this paper in the radiation scheme of a weather or climate model, these albedos would be used as the boundary conditions for the computation of the flux profile through the atmosphere.

### 2.6 Computing fluxes within the canopy

After running the atmospheric part of the radiation scheme, we proceed down through the vegetation to compute the direct and diffuse fluxes at each interface, ending up at the surface. The output from the atmospheric radiation calculation includes the downwelling direct and diffuse fluxes at the top of the canopy, $s_{1/2}$ and $v_{1/2}$. These are partitioned into component fluxes at the top of each region according to the area fraction of each region:

$$\mathbf{s}_{1/2} = s_{1/2}\mathbf{c}; \tag{37}$$
$$\mathbf{v}_{1/2} = v_{1/2}\mathbf{c}. \tag{38}$$

The direct flux is propagated down through the vegetation simply with

$$\mathbf{s}_{i+1/2} = \mathbf{E}_{0i}\mathbf{s}_{i-1/2}. \tag{39}$$

The diffuse fluxes at the interface beneath satisfy

$$\mathbf{u}_{i+1/2} = \mathbf{A}_{i+1/2}\mathbf{v}_{i+1/2} + \mathbf{D}_{i+1/2}\mathbf{s}_{i+1/2}; \tag{40}$$
$$\mathbf{v}_{i+1/2} = \mathbf{T}_i\mathbf{v}_{i-1/2} + \mathbf{R}_i\mathbf{u}_{i+1/2} + \mathbf{S}_i^-\mathbf{s}_{i-1/2}. \tag{41}$$



**Table 1.** Variables describing the geometry of 'Open forest' and 'Shrubland' RAMI4PILPS scenarios simulated in this paper (see Widlowski et al, 2011).

| Variable | Symbol | Open forest | Shrubland |
|---|---|---|---|
| Leaf Area Index of vegetated region | LAI | 5 | 2.5 |
| Area fraction of vegetated region | $c_v$ | 0.1, 0.3, 0.5 | 0.1, 0.2, 0.4 |
| Effective tree diameter | $D$ | 10 m | 1 m |
| Canopy layer depth | $\Delta z_1$ | 10 m | 1 m |
| Sub-canopy layer depth | $\Delta z_2$ | 4 m | 0.01 m |

**Table 2.** Variables describing the optical properties of the leaves and the surface in the visible and near-infrared in the RAMI4PILPS cases (see Widlowski et al, 2011).

| Variable | Symbol | Visible | Near-infrared |
|---|---|---|---|
| Leaf reflectance | $r$ | 0.0735 | 0.3912 |
| Leaf transmittance | $t$ | 0.0566 | 0.4146 |
| Snow-free surface albedo | $\alpha_{\mathrm{med}}$ | 0.1217 | 0.2142 |
| Snow albedo | $\alpha_{\mathrm{snow}}$ | 0.9640 | 0.5568 |

Eliminating $\mathbf{u}_{i+1/2}$ yields

$$\mathbf{v}_{i+1/2} = \left(\mathbf{I} - \mathbf{R}_i \mathbf{A}_{i+1/2}\right)^{-1}$$
$$\times \left(\mathbf{T}_i \mathbf{v}_{i-1/2} + \mathbf{R}_i \mathbf{D}_{i+1/2} \mathbf{s}_{i+1/2} + \mathbf{S}_i^- \mathbf{s}_{i-1/2}\right). \tag{42}$$

Thus, application of (42) followed by (40) provides the fluxes at the interface below.

5     The horizontally averaged upwelling diffuse, downwelling diffuse and downwelling direct fluxes at interface $i+1/2$, denoted $u_{i+1/2}$, $v_{i+1/2}$ and $s_{i+1/2}$, respectively, are found by simply summing the elements of $\mathbf{u}_{i+1/2}$, $\mathbf{v}_{i+1/2}$ and $\mathbf{s}_{i+1/2}$. The total downwelling flux is then the sum of the direct and diffuse components: $d_{i+1/2} = \mu_0 s_{i+1/2} + v_{i+1/2}$. The solar absorption by each layer is the difference in net flux between the interface above and below it. These definitions are used to compute normalized quantities that will be used to evaluate SPARTACUS in section 3.

10 **3 Results**

To test the application of the SPARTACUS methodology to the vegetation problem, we use two three-dimensional scenarios from the RAMI4PILPS[1] intercomparison exercise (Widlowski et al, 2011). The first scenario is an idealized representation

---

[1]RAMI is the Radiation Transfer Model Intercomparison, and PILPS is the Project for Intercomparison of Land surface Parameterization Schemes.





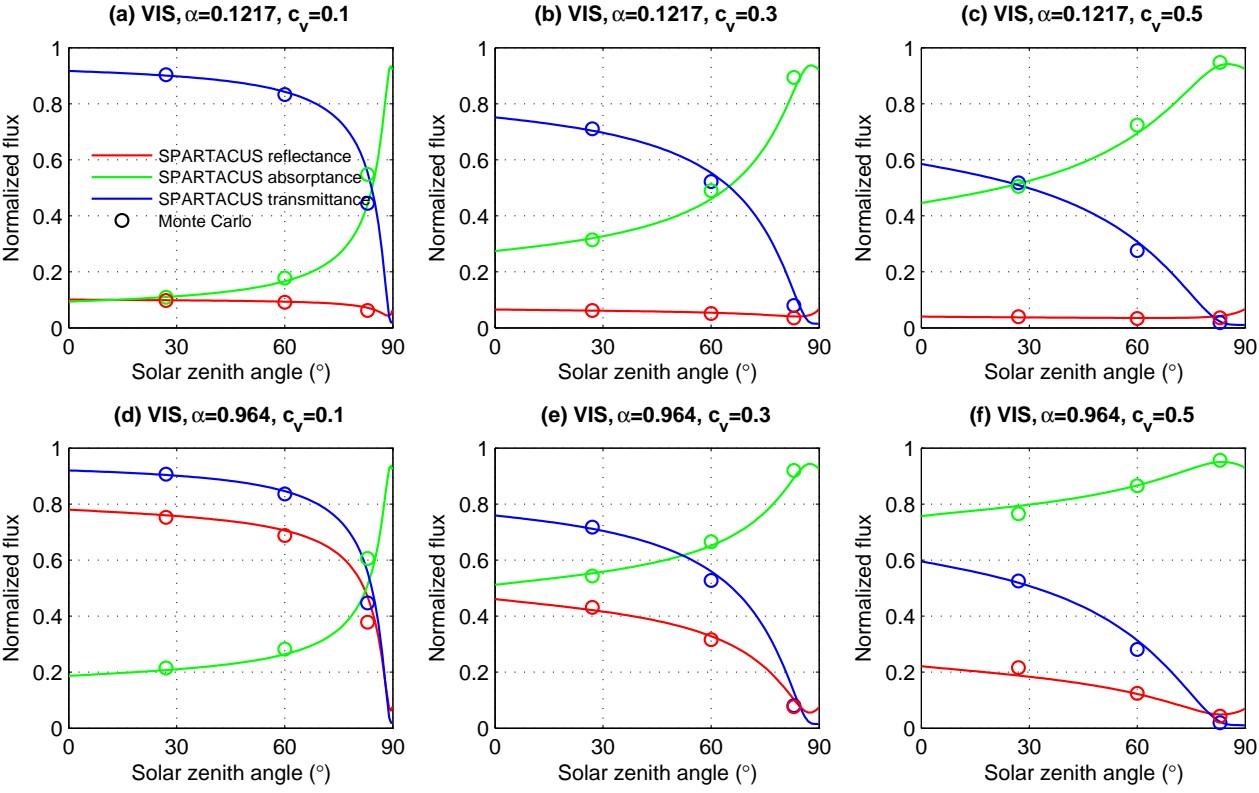

**Figure 2.** Comparison of normalized fluxes versus solar zenith angle for the RAMI4PILPS 'Open forest canopy' scenario with optical properties appropriate for visible radiation. The two rows of panels show results for different surface albedos ($\alpha$) with the top row using values appropriate for a snow-free surface and the bottom row using values for a snow-covered surface. The columns represent different areal tree fractions ($c_v$). The three lines depict the reflectance, transmittance and absorptance defined in (43), (44) and (45). The lines show 3-region SPARTACUS calculations, and are compared to Monte Carlo calculations at three solar zenith angles from Widlowski et al (2011).

of an open forest canopy, and consists of spheres of leafy vegetation of diameter 10 m with an average leaf area index of 5, while the second represents shrubland and consists of spheres of diameter 1 m with an average leaf area index of 2.5. Details are provided in Table 1, including the three different area coverages of vegetation that are used. Two spectral intervals are considered, representing the photosynthetically-active visible region and the near-infrared, and both snow-free and snow-covered surfaces are considered. Table 2 lists the optical properties of the leaves and the surfaces in the two spectral intervals.

All combinations have been simulated using the three-region ($m = 3$) version of SPARTACUS. The two vegetated regions ($b$ and $c$) are configured to approximate the distribution of zenith optical depth (or equivalently LAI) of spheres. So for a sphere with a mean LAI of 5, region $b$ represents the lower half of the distribution with leaf area index $\mathrm{LAI}^b = 0.5\,\mathrm{LAI}$, while region $c$ represents the upper half of the distribution and is assigned a leaf area index of $\mathrm{LAI}^c = 1.5\,\mathrm{LAI}$.





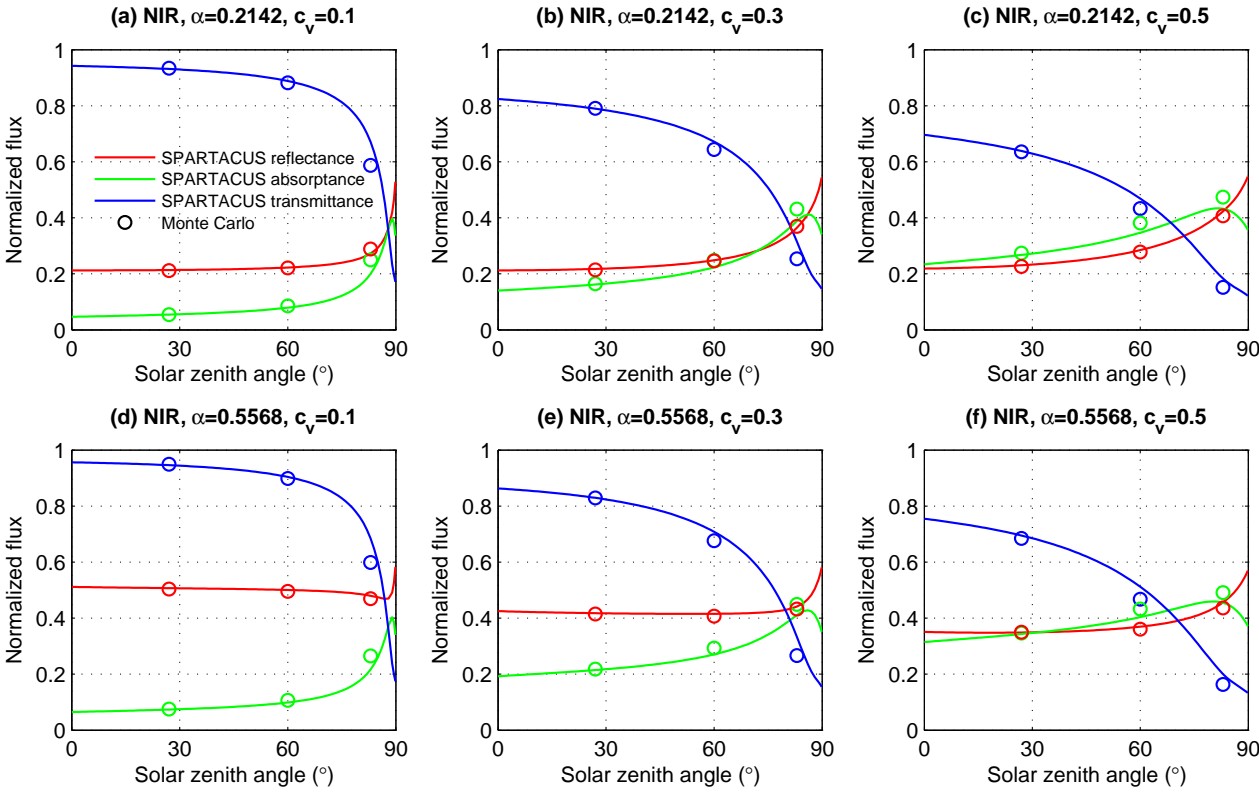

**Figure 3.** As Fig. 2 but with optical properties appropriate for near-infrared radiation.

Figure 2 shows the results for the open forest canopy in the visible part of the spectrum while Fig. 3 shows the same but for the near-infrared. Using the domain-mean fluxes defined in section 2.6, the quantities shown are reflectance $R$, transmittance $T$ and absorptance $A$:

$$R = u_{1/2}/d_{1/2};$$ (43)

$$T = d_{n+1/2}/d_{1/2};$$ (44)

$$A = \left(d_{1/2} - u_{1/2} - d_{n+1/2} + u_{n+1/2}\right)/d_{1/2}.$$ (45)

It can be seen that SPARTACUS performs well in all scenarios, including all four combinations of high- and low-reflectance leaves over a high- or low-reflectance surface. The performance is also good for the shrubland scenario shown in Figs. 4 and 5.

We next investigate how the results are degraded when using a more approximate description of the scene. The solid lines in Fig. 6 show the same 3-region SPARTACUS results as Figs. 2b and 2e, corresponding to the open forest canopy illuminated by visible radiation. Since individual trees have a leaf area index of 5 and the areal tree cover is $c_v = 0.3$, the domain-mean leaf area index is 1.5.





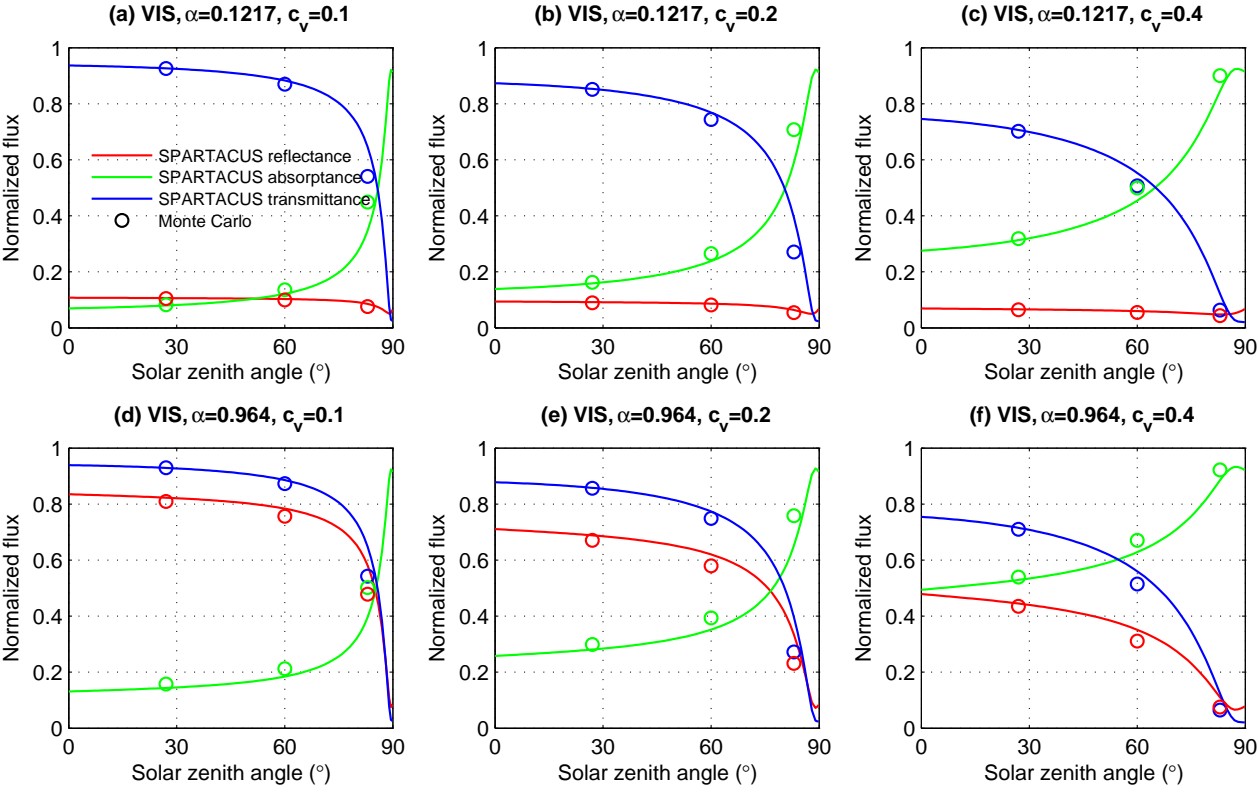

**Figure 4.** As Fig. 2 but for the RAMI4PILPS 'Shrubland' scenario.

Two additional lines are shown. The 1-region calculation treats the canopy as a single horizontally homogeneous layer with the same leaf area index. This is essentially the same as the Sellers (1985) assumption and indeed with a single region the matrix-exponential method yields the same result as the Meador and Weaver (1980) solution. We see immediately that when the leaves are not clumped into trees but rather distributed uniformly, their exposure to incoming radiation is maximized and their

5  absorptivity is overestimated by up to 0.3. Conversely, both the reflectance and transmittance of the scene are underestimated, with the largest error in reflectance for overhead sun and a snow-covered surface.

The 2-region SPARTACUS calculation treats individual trees as horizontally homogeneous cylinders, thereby neglecting the variation in zenith optical depth of the spherical trees simulated by the Monte Carlo calculations. The results are much better than those with just a single region, but solar absorption is still overestimated. An analogous bias occurs in cloudy radiative

10  transfer calculations in which the internal variability of clouds is neglected, leading to the proposal of Shonk and Hogan (2008) to use three regions to represent a partially cloudy scene. The success of the 3-region approach in Fig. 6 confirms that it is also applicable to vegetation. Having said this, the uncertainty in computing radiative transfer the vegetation canopies of weather and climate models is typically dominated by uncertainties in leaf area index. Therefore, for many applications the 2-region calculation would be adequate. Since the computational cost of SPARTACUS is dominated by the matrix exponential





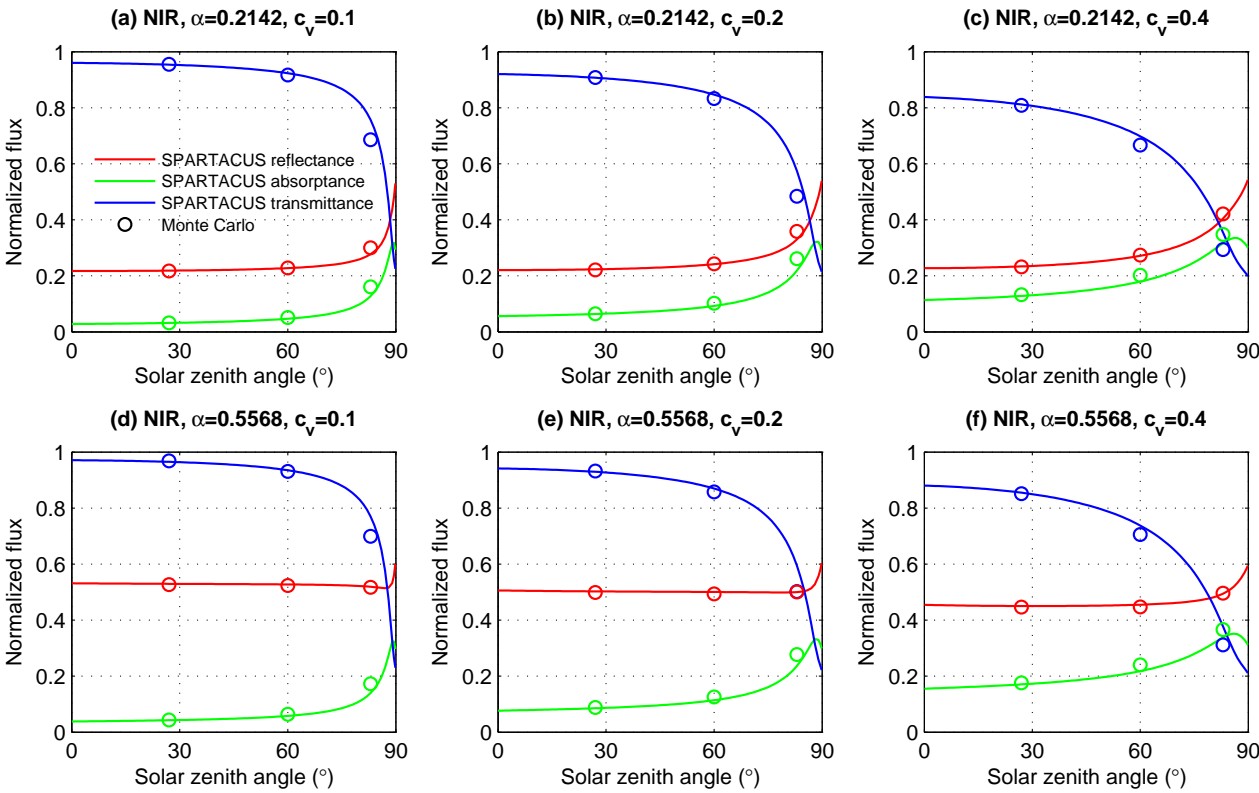

**Figure 5.** As Fig. 4 but with optical properties appropriate for near-infrared radiation.

calculation, whose cost is approximately proportional to $m^3$, we would expect a 2-region SPARTACUS calculation to be at least 3 times faster than a 3-region calculation.

## 4 Conclusions

This paper has demonstrated the potential for the interaction of solar radiation and complex vegetation canopies to be repre-
5 sented via an explicit description of the geometry, building on the SPARTACUS algorithm for representing the 3D radiative effects of clouds (Hogan et al., 2016). The two-stream equations are written down for the vegetation elements and the gaps between them, but with additional terms for the horizontal exchange of radiation between regions. The equations are solved exactly using the matrix exponential method. Multiple layers are possible, although we have simplified the original SPARTA-CUS algorithm by assuming maximum overlap between the regions in each layer, rather than the arbitrary overlap considered
10 by Hogan et al. (2016). Comparison against Monte Carlo calculations from the RAMI4PILPS intercomparison exercise indicates that canopy reflectance, transmittance and absorptance are computed to within 0.05 in the visible part of the spectrum, which is significantly better than a number of state-of-the-art models assessed by Loew et al. (2014).



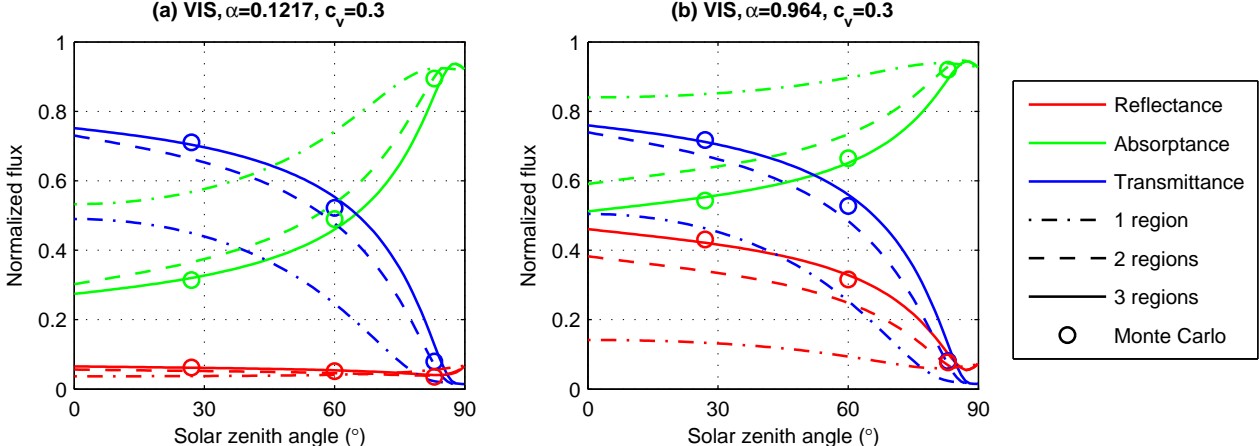

**Figure 6.** As Figs. 2b and 2e, but showing results from three SPARTACUS configurations: '1 region' homogenizes the vegetation horizontally through the entire domain and gives essentially the same result as the Sellers (1985) approach, '2 regions' homogenizes the vegetation within the individual tree crowns, while '3 regions' is the same as shown in Fig. 2 and depicted schematically in Fig. 1. Note that all three have the same domain-averaged leaf area index.

An advantage of the SPARTACUS approach is that in addition to LAI, only a handful of physiographic variables are required to describe the geometry of the vegetation, such as the vegetation height, coverage, and the diameter of typical vegetation elements. Global estimates of the first two are now available from satellites (e.g., Simard et al., 2011; Hansen et al., 2003).

Although the testing scenarios used in this papers were simple homogeneous spheres with no woody material, the method
described has the capability to represent more complex geometries. Horizontal variations in leaf density or vegetation elements with different properties may be represented via two or more vegetated regions with distinct optical properties. This paper considered a two-layer description of the vegetation, with a single canopy layer overlying an sub-canopy layer, but the equations can easily be applied to a multi-layer description of the canopy, for example to compute the vertical profile of absorbed photosynthetically active radiation. The optical effects of tree trunks may also be incorporated. Moreover, the good performance
with solar radiation suggests that the thermal-infrared version of SPARTACUS (Schäfer et al., 2016) could also be adapted to the vegetation problem.

**Code availability**

A Matlab implementation of the algorithm is freely available from http://www.met.reading.ac.uk/clouds/spartacus, and was used to produce Figs. 2–6. Work is in progress to implement the algorithm in the 'ecRad' atmospheric radiation scheme
(Hogan and Bozzo, 2016).



## Appendix A: Faster treatment of clear layers

The main role of the sub-canopy layer is to represent how much of the sunlight passing down between the trees is reflected back up into the base of a vegetation element, i.e. the off-diagonal elements of $\mathbf{A}_{n-1/2}$ and $\mathbf{D}_{n-1/2}$. Since the matrix exponential accounts for most of the cost of the scheme, if we can accelerate or approximate the treatment of the sub-canopy layer in a way

that avoids the full matrix-exponential calculation in this layer then we can almost halve the overall computational cost. This is only possible if we assume that the sub-canopy layer contains no absorbers or scatterers ($\sigma = \sigma_0 = 0$), i.e. tree trunks and understory vegetation are neglected.

There are two extreme scenarios that lead to $\mathbf{A}_{n-1/2}$ and $\mathbf{D}_{n-1/2}$ having trivial forms. For shrubs with a very shallow sub-canopy layer, the lateral transport between the regions of this layer is zero, leading to albedo matrices at the interface between

the canopy and sub-canopy layer being equal to the values at the surface given by (35) and (36). For a very deep sub-canopy layer, the radiation field beneath the canopy is randomized horizontally, leading to the diffuse albedo having the form

$$\mathbf{A}_{n-1/2} \simeq \begin{pmatrix} c^a & c^a & c^a \\ c^b & c^b & c^b \\ c^c & c^c & c^c \end{pmatrix} \overline{\alpha_{\mathrm{diff}}}, \tag{A1}$$

where $\overline{\alpha_{\mathrm{diff}}}$ is the domain-averaged surface albedo to diffuse radiation. The direct albedo $\mathbf{D}_{n-1/2}$ has a similar form.

For sub-canopy layers with a depth between these two extremes, we seek to optimize the calculation of the matrix expo-

nential. The lack of scattering means that the $\Gamma_2$, $\Gamma_3$ and $\Gamma_4$ sub-matrices contain only zeros, and $\Gamma$ becomes block-diagonal. This enables the exponential of a $3m \times 3m$ matrix to be replaced by three $m \times m$ matrix-exponential calculations, only two of which are needed: $\mathbf{E}_0 = \exp(\Gamma_0 \Delta z)$ and $\mathbf{E}_{vv} = \exp(\Gamma_1 \Delta z)$. Since there is no scattering in the sub-canopy layer, the matrices $\mathbf{R}$, $\mathbf{S}^+$ and $\mathbf{S}^-$ contain only zeros. Therefore, (27) simplifies to $\mathbf{T} = \mathbf{E}_{vv}$, and (33) and (34) simplify to

$$\mathbf{A}_{n-1/2} = \mathbf{T}_n \mathbf{A}_{n+1/2} \mathbf{T}_n; \tag{A2}$$

$$\mathbf{D}_{n-1/2} = \mathbf{T}_n \mathbf{D}_{n+1/2} \mathbf{E}_{0n}. \tag{A3}$$

Moreover, by approximating the extinction coefficients as zero, we see from (3) and (4) that $\Gamma_0$ and $\Gamma_1$ have simpler forms whose matrix exponentials can be derived analytically. In the $m = 2$ case these matrices have the form

$$\Gamma' = \begin{pmatrix} -a & b \\ a & -b \end{pmatrix}, \tag{A4}$$

for which the matrix exponential is given by Putzer's algorithm as

$$\exp\left(\Gamma' \Delta z\right) = \mathbf{I} + \frac{1 - \mathrm{e}^{-(a+b)\Delta z}}{a+b} \Gamma'. \tag{A5}$$

Likewise in the $m = 3$ case these matrices have the form

$$\Gamma' = \begin{pmatrix} -a & b & 0 \\ a & -b-c & c \\ 0 & c & -c \end{pmatrix}, \tag{A6}$$



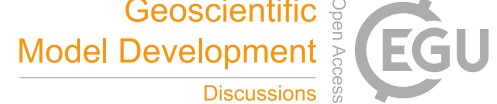

for which the matrix exponential may be computed by the diagonalization method as

$$\exp\left(\mathbf{\Gamma}'\Delta z\right) = \mathbf{V}\begin{pmatrix} e^{\lambda_1 \Delta z} & & \\ & e^{\lambda_2 \Delta z} & \\ & & 1 \end{pmatrix}\mathbf{V}^{-1}, \tag{A7}$$

where the two non-zero eigenvalues are

$$\lambda = -(a+b+2c)/2 \pm (a^2+b^2+4c^2+2ab-4ac)^{1/2}/2, \tag{A8}$$

5   and the matrix of eigenvectors is

$$\mathbf{V} = \begin{pmatrix} b/(a+\lambda_1) & b/(a+\lambda_2) & b/a \\ 1 & 1 & 1 \\ c/(c+\lambda_1) & c/(c+\lambda_2) & 1 \end{pmatrix}. \tag{A9}$$

*Acknowledgements.* We thank Jean-Luc Widlowski for providing the Monte Carlo results. TQ's contribution was funded by the UK National Centre for Earth Observation. RB was supported by a scholarship from the Brazilian 'Science without Borders' Program (grant number 9549-13-7), financed by CAPES, the Brazilian Federal Agency for Support and Evaluation of Graduate Education within the Ministry of Education of Brazil.





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
