# Peer review of "Fast matrix treatment of 3D radiative transfer in vegetation canopies: SPARTACUS-Vegetation 1.1"

_Geoscientific Model Development, 2017_

## Short Comment (SC1) · 10 Oct 2017

Robin

As explained in https://www.geoscientific-model-development.net/about/manuscript_types.html GMD is encouraging authors to upload the program code of models as a supplement in order to maintain reproducibility of the results presented in the paper. I would like to encourage you to do this for the matlab code you have used to create the results in the final version of your paper. Please feel free to point the reader in the Code Accessibility section to http://www.met.reading.ac.uk/clouds/spartacus to access updated version.

[Figure]

Many thanks.

Lutz Gross GMD Executive Editor

---

## Referee Comment (RC1) · Anonymous Referee #1 · 19 Oct 2017

The manuscript describes a new approach to quickly and accurately calculate the spectral albedo of a vegetation canopy using information on its optical and structural properties. The presented algorithm is a clear advancement compared with the dominating 2-stream approximation as it represent realistic forests and shrublands as consisting of individual trees. The paper addresses a relevant issue and has imminent application potential. The description of the method is sufficiently complete and precise. However, some minor technical modifications are needed for greater clarity.

Minor (technical) comments: Page 1 line 1: I suggest specifying the (geographic) scale where the model can be applied and also the scale of the "regions". line 2: Splitting

"horizontally" is ambiguous. It can be understood as splitting with horizontal planes. I suggest using e.g. "split in the horizontal plane". line 7: I suggest adding some quantitative results, e.g. the number 0.05 mentioned in Conclusions.

Page 2: The description of clumping is misleading. Clumping is also used to describe vegetation structure variation in the vertical direction and at scales smaller than a tree crown. This should be mentioned as the current description can be misleading with respect to the universality of the proposed approach.

Page 3 line 4: add "constant thickness" after "canopy layer". I see no need for quatition marks. line 4: Define what is meant by "domain" line 4: Again, choose an anambigous term instead of "divided horizontally" (although the meaning can be inferred from context). line 4: The concept of "region" should be defined here and not on the following page. It is counterintuitive to have a region consisting of separate parts. line 4: The necessity of up to two vegetated regions is not justified and not followed later in the manuscript. line 7: How would the situation of objects not being cylinders (highly grouped canopies) affect accuracy? In my opinion, this is explicitly assumed here. Although not mathematically, but the results are only provided for canopies with clearly separable crowns. line 8: Define "vegetation element". E.g., is it a leaf or a tree crown? line 9: Unclear what is meant by "same": a canopy layer is first and foremost defined by leaf area density. line 10: Why possible omission is only mentioned for shrubland?

Page 4 line 1: Define what a,b,c stand for (different regions). Probably, it needs to be done earlier as line 13 of previous page already refers to $L^{ab}$. In hindsight, it is clear that a and b refer to two regions. line 1: In optical radiometry, radiant power is the same as radiant flux. Use only one of these terms consistently. FLux per surface area (flux density) is irradiance. "Domain-mean" flux is a contradictory term. Irradiance can be averaged, but flux being total power can only be added. The correct term would be domain-total flux, the sum all flux components over the domain. (note: in many other fields, flux is power divided by area) line 3: This line contains the definition of a "region". It should be given earlier.

Page 5 line 25: rewrite as LAI/(2 Delta z) line 32: citation needed for the equations.

Page 6 line 8: Give some justifications for this rather arbitrary assumption and also discuss its consequences. line 12: Unclear what is meant by "random" and why it's necessary. Different random processes can dreate very different tree distribution patterns, but very few create non-overlapping crowns. Instead of "random", why cannot the trees in the "idealized forest" be situated on a regular grid?

Page 11 line 4: Choose either PAR or "visible region"; alternatively, add "and" between the two. line 7: Be moerelaborate on the approximation method. line 7: A sphere (or, a single tree crown) does have a LAI value. LAI is only defined for a region which usually includes betweem-element gaps, e.g., a forest stand. It can indeed be defined for the area of a single crown, but this contradicts the common practice. line 8: Clarify what is meant by "upper" and "lower". These do not seem to refer to canopy location (but can be understood to).

Page 12 line 2: Again, "domain-main flux" needs clarification. line 11: Again, I suggest avoiding the use of LAI for a single tree. It is straightforward for ideal cylindrical tree crowns, but can cause much confusion when attempted in a natural situation where tree crowns do not have a clearly distinguishable bounding surface.

Page 14 The section "Conclusions" contains mostly discussion and should be re-named. No new issues should be brought up in Conclusions and citations are unnecessary. Instead, the statements should be based on what was presented earlier, mainly Discussion – a section clearly missing from the manuscript. The current Conclusions contains many new topics and even a value (0.05 on line 11, which should be mentioned in the results section).

―――――――――――――――

---

## Referee Comment (RC2) · Anonymous Referee #2 · 21 Oct 2017

The modelling of radiative transfer in plant canopies is an important aspect of the land surface component of weather forecasting and climate models, with impacts both directly on the radiation field and indirectly on photosynthesis. An accurate and reliable scheme is therefore required; but typically those employed in such models are relatively simplistic. The authors describe a new scheme that takes better account of horizontal heterogeneity in vegetative canopies and test it against two benchmarks, finding significant improvements from the representation of heterogeneity and the associated lateral transport of radiation. The paper is well written, scientifically sound and certainly within the scope of GMD. I recommend that it should be accepted subject to minor revision. Specific comments follow.

[Figure]

1.) Page 3, line 7. Please add "ly" after "explicit."

2.) Page 6, lines 10–21. I am unclear about the fundamental parameter here. The equations require $L^{ab}$. Is this what you would measure in the field, or would you measure $D$ and infer $L^{ab}$? In the former case, $D$ is just an illustrative diameter, but is more fundamental in the latter case. In the case of dense canopies, if $L^{ab}$ is measured, what is the purpose of $S$, the meaning of which is unclear? Conversely, if you infer $L^{ab}$ from $S$, how is $S$ determined in the field?

3.) Page 11, lines 6–9. I assume that regions $b$ and $c$ still have the same area, as noted on page 6. It would be useful to remind the reader of this. On line 8, the argument should apply to any sphere, not just one with an LAI of 5. It is not clear to me why factors of 0.5 and 1.5 have been chosen. If the distribution of zenith optical depth is split into two equal parts by projected area, I expect the denser region to correspond to a core of radius $r/\sqrt{2}$ excised from a sphere of radius $r$. In this case I think the core will contain about 65% of the volume of the sphere and so the same fraction of the total leaf area. I would therefore expect the proportions to be 0.707 and 1.293, not 0.5 and 1.5.

4.) Figure 6. Previously, results for both the VIS and NIR regions have been shown. Why is the NIR omitted here? Unless the differences are trivial I would suggest showing this region too.

5.) The authors note (page 13, line 12) that there are large uncertainties in the LAI used in weather and climate models. The underlying datasets are derived from remote sensing, so it would be interesting if the authors could comment on the possible application of their model in the retrieval of LAI. The use of a consistent modelling framework in these two areas would be of considerable value.

---

## Author Comment (AC1) · 6 Dec 2017

Response to referees' comments: please see revised draft of manuscript (included as a supplement) with changes marked in red.

Referee 1

COMMENT 1. Page 1 line 1: I suggest specifying the (geographic) scale where the model can be applied and also the scale of the "regions". line 2: Splitting "horizontally" is ambiguous. It can be understood as splitting with horizontal planes. I suggest using e.g. "split in the horizontal plane". line 7: I suggest adding some quantitative results,

e.g. the number 0.05 mentioned in Conclusions.

REPLY 1. The first line now says "vegetation canopies" rather than "vegetation" to make clear that radiation is treated at a larger scale than individual trees. The final sentence talks about weather and climate modelling, so the scale is then clearly the size of a gridbox of such models. The phrase "split in the horizontal plane" is now used, and the root-mean-squared differences in reflectance, transmittance and canopy absorptance have now been stated.

COMMENT 2. Page 2: The description of clumping is misleading. Clumping is also used to describe vegetation structure variation in the vertical direction and at scales smaller than a tree crown. This should be mentioned as the current description can be misleading with respect to the universality of the proposed approach.

REPLY 2. The text has been changed accordingly.

COMMENT 3. Page 3 line 4: add "constant thickness" after "canopy layer". I see no need for quatition marks.

REPLY 3. "Constant thickness" has been added. The quotation marks are to clarify that "canopy layer" and "sub-canopy layer" are named layers in the formulation of the problem, as shown in Fig. 1.

COMMENT 4. ...line 4: Define what is meant by "domain" line 4: Again, choose an anambigous term instead of "divided horizontally" (although the meaning can be inferred from context). line 4: The concept of "region" should be defined here and not on the following page. It is counterintuitive to have a region consisting of separate parts.

REPLY 4. I've expanded to "horizontal domain (corresponding to a weather- or climate-model gridbox)". The definition of region has been improved. In no case in this paper does a region consist of separate parts. But a tree can be represented by separate regions (as shown in Fig. 1).

COMMENT 5. ...line 4: The necessity of up to two vegetated regions is not justified

and not followed later in the manuscript.

REPLY 5. The need for two vegetated regions is justified in Fig. 6 of the original manuscript: when only one vegetated region is used, a worse result is obtained. In the new manuscript, the results for one vegetated region are added to Figs. 2-5. Section 2.1 now makes reference to these results in section 3.

COMMENT 6. ...line 7: How would the situation of objects not being cylinders (highly grouped canopies) affect accuracy? In my opinion, this is explicitly assumed here. Although not mathematically, but the results are only provided for canopies with clearly separable crowns.

REPLY 6. The new text at the end of section 2.3 explains how the assumptions in SPARTACUS are only that the vegetation is randomly separated, not that it is composed of cylindrical crowns. However, it will need to wait until a future paper to test this since we only have Monte Carlo results from RAMI4PILPS where the nature of the canopy is rather geometrically simple.

COMMENT 7. ...line 8: Define "vegetation element". E.g., is it a leaf or a tree crown? line 9: Unclear what is meant by "same": a canopy layer is first and foremost defined by leaf area density. line 10: Why possible omission is only mentioned for shrubland?

REPLY 7. These terms have been clarified: "vegetation element" replaced by "tree crown", and the phrase containing "same" replaced with "also divided into m regions (see Fig. 1)". The mention of shrubland has been removed.

COMMENT 8. Page 4 line 1: Define what a,b,c stand for (different regions). Probably, it needs to be done earlier as line 13 of previous page already refers to $L^{ab}$. In hindsight, it is clear that a and b refer to two regions.

REPLY 8. Now defined in section 2.1.

COMMENT 9. ...line 1: In optical radiometry, radiant power is the same as radiant flux. Use only one of these terms consistently. FLux per surface area (flux density) is

irradiance. "Domain-mean" flux is a contradictory term. Irradiance can be averaged, but flux being total power can only be added. The correct term would be domain-total flux, the sum all flux components over the domain. (note: in many other fields, flux is power divided by area)

REPLY 9. All uses of flux where such an ambiguity can arise have been replaced by irradiance.

COMMENT 10. ...line 3: This line contains the definition of a "region". It should be given earlier.

REPLY 10. It is now given in section 2.1.

COMMENT 11. Page 5 line 25: rewrite as LAI/(2 Delta z)

REPLY 11. Done.

COMMENT 12. line 32: citation needed for the equations.

REPLY 12. Citation provided (Pinty et al. 2006).

COMMENT 13. Page 6 line 8: Give some justifications for this rather arbitrary assumption and also discuss its consequences.

REPLY 13. The justification is that it was found by Shonk & Hogan (2008) to be the best assumption for representing the PDF of cloud optical depth, as is stated. Further justification is provided by the good a-posteriori agreement with Monte Carlo, shown in the results section.

COMMENT 14. line 12: Unclear what is meant by "random" and why it's necessary. Different random processes can dreate very different tree distribution patterns, but very few create non-overlapping crowns. Instead of "random", why cannot the trees in the "idealized forest" be situated on a regular grid?

REPLY 14. We have now explaned mathematically at the end of section 2.3 how the

SPARTACUS formulation implies a random distribution of trees. Specifically, it assumes that the chord lengths between the edges of tree crowns in all possible horizontal directions follow an exponential distribution. If the trees were on a regular grid, their spacing distribution would be far from exponential, and the assumption would be violated.

COMMENT 15. Page 11 line 4: Choose either PAR or "visible region"; alternatively, add "and" between the two.

REPLY 15. We believe that the text reads better if it is kept the same: the spectral interval from 400 to 700 nm is both photosynthetically active and the range detectable by the human eye. Adding an "and" would make the phrase more confusing when it is immediately followed by "and the near-infrared".

COMMENT 16. ...line 7: Be moerelaborate on the approximation method.

REPLY 16. The explanatory text that follows has been expanded - see also the reply to Referee 2's Comment 3.

COMMENT 17. ...line 7: A sphere (or, a single tree crown) does have a LAI value. LAI is only defined for a region which usually includes betweem-element gaps, e.g., a forest stand. It can indeed be defined for the area of a single crown, but this contradicts the common practice. line 8: Clarify what is meant by "upper" and "lower". These do not seem to refer to canopy location (but can be understood to).

REPLY 17. Where possible, such discussion is rephrased in terms of zenith optical depth. However, Widlowski et al. (2011) did use LAI in this context in their Table 1, so we do too in our Table 1 but with more explanation. "Upper" and "lower" refer to parts of the zenith optical depth distribution, not vertical location. We have tried to make this clearer.

COMMENT 18. Page 12 line 2: Again, "domain-main flux" needs clarification. line 11: Again, I suggest avoiding the use of LAI for a single tree. It is straightforward for ideal cylindrical tree crowns, but can cause much confusion when attempted in a natural

situation where tree crowns do not have a clearly distinguishable bounding surface.

REPLY 17. Rephrased.

COMMENT 18. Page 14 The section "Conclusions" contains mostly discussion and should be renamed. No new issues should be brought up in Conclusions and citations are unnecessary. Instead, the statements should be based on what was presented earlier, mainly Discussion – a section clearly missing from the manuscript. The current Conclusions contains many new topics and even a value (0.05 on line 11, which should be mentioned in the results section).

REPLY 18. Root-mean-squared errors are now computed and stated in the results section. The final section has been renamed "Discussion and conclusions"

Referee 2

COMMENT 1. Page 3, line 7. Please add "ly" after "explicit."

REPLY 1. Done

COMMENT 2. Page 6, lines 10–21. I am unclear about the fundamental parameter here. The equations require $\hat{L}^{ab}$. Is this what you would measure in the field, or would you measure D and infer $\hat{L}^{ab}$? In the former case, D is just an illustrative diameter, but is more fundamental in the latter case. In the case of dense canopies, if $\hat{L}^{ab}$ is measured, what is the purpose of S, the meaning of which is unclear? Conversely, if you infer $\hat{L}^{ab}$ from S, how is S determined in the field?

REPLY 2. The fundamental parameter for 3D radiation is $\hat{L}^{ab}$. However, this depends on both the areal coverage of trees "$c\_v$", and the properties of an individual representative tree. In the context of a weather or climate simulation, we would use a global dataset of $c\_v$ (e.g. from Hansen et al) but would need to estimate $\hat{L}^{ab}$ from it. This can be done by introducing an additional parameter representing the size of an individual tree, and the manuscript describes two models for how this could be done (D and S); to be useful, the parameter used would need to be independent of $c\_v$. To com-

pute D and S in the field, we would measure Lˆab and c_v and apply inverted forms of equations 17 and 18. D is needed for comparison with the Monte Carlo results in the present manuscript which assumed tree crowns not to touch. The manuscript has been extended to clarify all these points in new section 2.4.

COMMENT 3. Page 11, lines 6–9. I assume that regions b and c still have the same area, as noted on page 6. It would be useful to remind the reader of this. On line 8, the argument should apply to any sphere, not just one with an LAI of 5. It is not clear to me why factors of 0.5 and 1.5 have been chosen. If the distribution of zenith optical depth is split into two equal parts by projected area, I expect the denser region to correspond to a core of radius r/$\sqrt{}$2 excised from a sphere of radius r. In this case I think the core will contain about 65% of the volume of the sphere and so the same fraction of the total leaf area. I would therefore expect the proportions to be 0.707 and 1.293, not 0.5 and 1.5.

REPLY 3. We have now reminded the reader that b and c have the same area, and removed the implication that the following argument works only for an LAI of 5. The reviewer is right that the factors have been computed incorrectly (thank you!). The new factors are now used in the paper, which changes the lines of the figures slightly.

COMMENT 4. Figure 6. Previously, results for both the VIS and NIR regions have been shown. Why is the NIR omitted here? Unless the differences are trivial I would suggest showing this region too.

REPLY 4: We have now added the 2-region and 1-region lines to Figs. 2-5 so the reader can see the effect in all cases, including NIR. This means that Fig. 6 is no longer needed.

COMMENT 5. The authors note (page 13, line 12) that there are large uncertainties in the LAI used in weather and climate models. The underlying datasets are derived from remote sensing, so it would be interesting if the authors could comment on the possible application of their model in the retrieval of LAI. The use of a consistent modelling

framework in these two areas would be of considerable value.

REPLY 5. This is now discussed in the final section.

Please also note the supplement to this comment:
https://www.geosci-model-dev-discuss.net/gmd-2017-208/gmd-2017-208-AC1-supplement.pdf

**Supplement:**

[revised manuscript text omitted]

---

## Author Comment (AC2) · 6 Dec 2017

We would be happy to upload the Matlab code as a supplement, and will do so when we have the opportunity to revise the manuscript.